# Occurrence and Characterization of Microplastics in Commercial Mussels (*Mytilus galloprovincialis*) from Apulia Region (Italy)

**DOI:** 10.3390/foods12071495

**Published:** 2023-04-02

**Authors:** Angela Dambrosio, Stefania Cometa, Flavia Capuozzo, Edmondo Ceci, Michele Derosa, Nicoletta Cristiana Quaglia

**Affiliations:** 1Department of Veterinary Medicine, University of Bari Aldo Moro, 70010 Valenzano, Italy; 2Jaber Innovation s.r.l., Via Calcutta 8, 00144 Rome, Italy; 3Veterinary Surgeon, Freelance Professional, Via Martiri di via Fani, 62, Grumo Appula, 70025 Bari, Italy

**Keywords:** microplastics, mussels, FTIR-ATR, food safety

## Abstract

Microplastics are a ubiquitous pollutant whose spreading is a growing concern worldwide. They can pose a threat to food safety and consumer health as they are ingested through various foods. Bivalves are considered the most contaminated, as they filter large amounts of seawater and enter consumers’ diet ingested whole. The aim of this study was to detect, quantify, identify and classify microplastics in mussels (*Mytilus galloprovincialis*) marketed in fishery stores in Bari and its surroundings (Apulia, Italy). A total of 5077 particles were isolated from our samples, with an average value of 1.59 ± 0.95 MPs/g and 6.51 ± 4.32 MPs/individual. Blue fragments, sized 10–500 µm, were the prevalent findings; most of them belonged to Polyamide (PA) polymers. The results of this study help to show that mussels represent a source of microplastics for consumers and a direct risk to their health, even considering that they may contain many chemical compounds and microorganisms that may or may not be pathogenic to humans. Further research is needed to assess the role of commercialization in bivalve molluscs contamination.

## 1. Introduction

Marine pollution by microplastics is a worldwide concern in terms of marine ecosystems and human health. Microplastics are found throughout the marine environment, from beaches and coasts, to the depths of the oceans, and were recently sampled in freshwater systems [1,2]. There is no internationally accepted definition of microplastics. However, in 2008, in order to standardize the definition of microplastics, the first “International Research Workshop on the Occurrence, Effects and Fate of Microplastic Marine Debris” proposed to consider an upper size limit, defining microplastics as “plastic particles smaller than 5 mm” [3]. This definition was further revised in 2011, when Cole et al. [4] distinguished into primary and secondary microplastics according to their origin. Primary microplastics are intentionally produced microscopic in size for direct use or as precursors to other products, including preproduction resin pellets, microbeads in cosmetics, toothpaste and blasting, powders for textile coatings, and drug delivery media. Secondary microplastics are tiny plastic fragments derived from the breakdown of larger plastic debris, both at sea and on land, including plastic fragments, microfibres from fabric and rope, coatings, and debris from tire wear [5].

Several types of plastics are produced worldwide, but the most common are classified into six classes of polymers: polyethylene (PE, high and low density), polypropylene (PP), polyvinyl chloride (PVC), polystyrene (PS), including expanded polystyrene (EPS), polyurethane (PUR) and polyethylene terephthalate (PET) [6]. They have many applications and can enter the waste stream quickly, such as packaging materials, or more slowly, such as plastics used in construction.

Among plastic waste, microplastics are of particular concern in terms of environmental pollution and, consequently, human and animal health, mainly because of their small size, availability and potential to be harmful to marine biota and humans [7]. Microplastics are highly persistent in the environment and, therefore, have been increasingly accumulating in various marine ecosystems [8].

Microplastics can be absorbed by a wide variety of marine organisms at different phases of their lives [1,9]. Of these, ingestion is probably the main route of exposure to microplastics [7]. In some cases, microplastics are ingested because they are confused with prey, but also through filter feeding [10]. After ingestion, microplastics can be absorbed and distributed through the circulatory system into different tissues and cells, potentially causing adverse effects [6]. Furthermore, microplastics, as well as the chemicals they contain [11], can be transferred from marine prey to predators [12]. Species that ingest microplastics include fish, crustaceans and seafood for human consumption. They represent a route of exposure for humans with health implications that are not fully understood yet [1].

To date, the presence of microplastics was observed in a wide range of commercially valuable fish products for human consumption, including fish (e.g., Atlantic cod, Atlantic horse mackerel; European sardine, mullet, European sea bass), bivalve molluscs (e.g., mussels, oysters) and crustaceans (e.g., brown shrimp) [13,14,15,16].

Fish products from aquaculture can also ingest microplastics [17]. In fact, aquaculture systems may involve feeding fish with fishmeal-based feed that is itself contaminated with microplastics present in raw materials [9].

Among all fish products, mussels were used as an indicator of marine environments contamination in biomonitoring programs [1]. This role is due to several important characteristics such as their wide geographical distribution, easy accessibility and high tolerance to a considerable range of salinities. Because they are able to filter large volumes of water, bivalves in particular were identified as species susceptible to microplastic ingestion and accumulation [18]. Consequently, the assessment of the presence of microplastics in mussels was proposed as a marine health parameter and included in the European database on environmental contaminants of emerging concern in seafood [19]. Mussels are, thus, both susceptible to microplastic contamination and a vector for microplastic into the human food chain.

In the Mediterranean Sea and in its fishery products, the occurrence of MPs was detected from Greek waters in the northern Ionian Sea, to central Adriatic coast, in Ligurian Sea and in Tyrrhenian Sea [17,20,21].

The European Union is the second largest producer of mussels with 522,000 tons per year. In 2016, Chinese and EU production accounted for 67% of global mussels’ production. In 2016, production in Italy, Denmark and Germany ranged from 44,000 to 64,000 tons. Together, these three Member States cover 28% of the total EU production [22]. In 2017, Italian family consumption of mussels amounted to 42,750 tons with a value of 102 million euros. In southern Italy, consumption of mussels is particularly high, as they are part of many traditional gastronomic preparations. Therefore, considering the high consumption of mussels, the health risk for consumers cannot be excluded.

The aim of this study was to evaluate the occurrence of microplastics in mussel samples collected from local fish markets. The microplastic debris were counted and the chemical identity confirmed using Fourier-transform infrared spectroscopy (FTIR).

## 2. Materials and Methods

### 2.1. Sample Collection and Preparation

The mussels (*M. galloprovincialis*) were purchased at 13 different fish markets in Bari and surrounding cities (Apulia region, Italy). The mussels were live, refrigerated and labelled in net bags, or unpackaged. Information on the declared species, the catch location of the wild mussels and the origin of the farmed mussels were obtained from the labels. Each 1 kg sample (from SM1 to SM13) consisted of approximately 60 individuals. To avoid contamination with microplastics due to transportation or other environmental factors, the collected samples were wrapped in aluminium foil and brought to the laboratory in a refrigerated box at 4 °C, where they were immediately analyzed. Each mussel sample was divided into three aliquots of 20 individuals each. Analyses were repeated on each aliquot, allowing a total of three replicates per sample.

### 2.2. Quality Control

To avoid contamination, all fluids (distilled water, saline solution and hydrogen peroxide) were filtered before use with a cellulose nitrate filter membrane with a pore size of 1 µm and a diameter of 47 mm (Axiva Sichem Biotech, Delhi, India). All equipment, containers and beakers were rinsed three times with filtered distilled water before and after use and covered with aluminum foil to prevent contamination by airborne microplastics.

At the same time, a blank extraction sample without tissue was performed to determine and correct any procedural contamination.

### 2.3. Hydrogen Peroxide Treatment

Each mussel was rinsed 3 times with pre-filtered distilled water to remove sediment and debris. The shell length and weight of each mussel was first recorded and then, thoroughly rinsed with filtered distilled water.

The mussels from each aliquot were opened, and the tissues from 5 individuals were pooled. A total of 10 g of the pool was placed in a 1 L glass bottle, and 200 mL of 30% H_2_O_2_ (1:20 *w*/*v*) (Honeywell/Fluka, Charlotte, NC, USA) was added to digest the organic matter. Each aliquot is considered a replicate; 3 replicates were prepared for each sample. The bottles were covered with aluminium foil and placed in an oscillation incubator at 65 °C 80 rpm for 24 h, then at room temperature for 24 or 48 h, depending on the digestion effect of the soft tissue. When no visible organic residues were left and the solution was clear, the digestion was considered completed [18].

### 2.4. Floatation and Filtration

A supersaturated NaCl solution (1.2 g mL^−1^) was used to separate the microplastics from the dissolved liquid of the soft tissue by floatation [18]. Approximately 800 mL of filtered NaCl solution was added to each bottle, mixed and incubated overnight at room temperature. The supernatant water was filtered through a cellulose nitrate membrane filter with 5 µm pore size and 47 mm diameter (Axiva Sichem Biotech, Delhi, INDIA) using Membrane–Laborpunpe (KNF Flodos AG, Sursee, Switzerland) under vacuum system. Extraction of the supernatant was performed in several sequential steps to maximize the recovery of MPs. The filters were placed in glass Petri dishes, covered, and dried at room temperature.

### 2.5. Observation of the Filters and Detection of the Items

The filters were observed under a stereomicroscope (Nikon, Italy) to analyse the presence of potential plastic particles and images were captured with a digital camera (Nikon X_Entry, Tokyo, Japan). The number, size, shape and colour of the debris on the filters were recorded [18]. The length of the detected particles was determined, and each particle was assigned to one of the four distinct size classes: 10–500 µm, 501–1000 µm, 1001–5000 µm and >5001 µm. The Nikon’s software for the imaging analysis was applied to the litter dimensional measurements (Nikon X_Entry, Tokyo, Japan).

### 2.6. FTIR/ATR Polymer Identification

The chemical characterization of plastic polymers was performed using Fourier-transform infrared spectroscopy in attenuated total reflectance mode (FTIR-ATR). The FTIR spectra of the samples were obtained with Spectrum Two PE instrument (PerkinElmer) equipped with the Universal ATR (UATR, Single Reflection Diamond/ZnSe) accessory and spectra were acquired in the range 400 to 4000 cm^−1^. The measurement resolution was set at 4 cm^−1^ with 16 scans. The analysis was performed on the microplastics removed from the membrane filters or straight on the filter. The particles were identified by comparing FT-IR absorbance spectra of the analysed MPs to those in a polymer reference library. An index of at least 70% match was considered acceptable [23].

### 2.7. Statistical Analysis

Data were analysed using Microsoft Excel software (Windows, 2010). Results were expressed as number of microplastics (MPs), number of microplastics/g (MPs/g) and number of microplastics/individual (MPs/individual). Mean values and standard deviation were calculated. One-way ANOVA (*p* < 0.05) was performed on the data expressed as MPs to detect significant differences between the number of MPs in samples and between the three replicates. Furthermore, to assess significant differences between the total number of microplastics in the samples and the total number of microplastics in the procedural blank (negative controls), the t-test (*p* < 0.001) was also performed.

## 3. Results

### 3.1. Abundance of Detected Items

According to the information on the labels, the mussels came from Italy and Greece (FAO zone 37.2).

All thirteen analysed samples were positive for the presence of plastic debris (Table 1).

The contamination from airborne microplastics in the blank samples was low, with an average value of 0.35 ± 0.17 compared to the maximum value of sample SM11 (3.62 ± 0.88 MPs/g). Plastic debris found in the blank samples were eliminated from the total count (Table 1).

A total of 5077 microplastics were isolated, with the lowest value in SM9 (108 MPs) and the highest value in SM11 (924 MPs). Three replicates of 20 individuals were performed for each sample; similar numbers of plastic elements were detected in all replicates; the average value of MPs in all samples was 1.59 ± 0.95 MPs/g or 6.51 ± 4.32 MPs/individual. The average value of each sample ranged from 0.53 ± 0.21 to 3.62 ± 0.88 MPs/g and from 1.8 ± 0.74 to 15.4 ± 3.75 MPs/individual (minimum and maximum value, respectively) (Table 1).

### 3.2. Statistical Analysis Results

Analysis of the data with the *t*-test showed that the number of microplastics in the samples was significantly higher than in the procedural blanks (*p* < 0.001). Statistical analysis (ANOVA) revealed a significant difference between the number of MPs in the samples (*p* < 0.05), while no significant difference was found between the results of the three replicates (*p* > 0.05).

### 3.3. Shape, Colour and Size of Detected Items

After morphometric analysis, four types of microplastics were observed: fibres, fragments, films and spherical granules. Fragments were the most commonly observed MPs (71.89%); fibres and films were detected in 23.1% and 4.89% respectively, and 0.85% of the MPs were spherical granules (Figure 1 and Figure 2).

The observed plastic debris had different colours: blue microplastics had a higher incidence (46.23%) than white MPs (0.57%) (Figure 1).

12.66% of the MPs was brown, while other colours, such as yellow (11.74%), transparent (10.95%), black (7.76%) grey (5.81%), green and red were less represented (Figure 1).

The size of microplastics ranged from 10 µm to 9080 µm in length, with an average value of 448.80 ± 527.66 µm in all mussels tested. The average value of fibre length was 1225.36 ± 396.31 µm, of fragments was 185.16 ± 287.79 µm, of films was 317.23 ± 357.57 µm and of spherical granules was 67.44 ± 115.75 µm. The most abundant microplastics were in the 10–500 µm size class, accounting for 80.72% of the total. A total of 9.57% of the MPs belonged to the 501–1000 µm size class and 1.18% to the >5000 µm size class (Figure 1).

### 3.4. Polymer Composition of the Microplastics

Through the analysis of the isolated MPs with the FTIR- ATR, different types of polymers were identified, including PA (68.73%), Polyvinyl alkyl carbamate (PVAC) (11.53%) with scores between 0.70 and 0.82, PUR (11.53%) and polypropylene (PP) (7.69%) (Table 2). Figure 3 shows the representative spectra obtained from the extracted MPs.

## 4. Discussion

### 4.1. Abundance of MPs in Mussels from Markets

Marine microplastics are an emerging concern for food security, food safety and human health and is a very important field of research. The presence of microplastics in fish species of commercial interest is a global problem and humans are vulnerable to microplastics through the consumption of seafood and other food. In particular, mussels are the most exposed to contamination with MPs.

In our survey, we investigated the occurrence of MPs in mussels (*M. galloprovincialis*) collected from different fish markets in Bari and neighbouring cities (Apulia region, Italy), to assess the direct risk to consumers.

Several studies assessed the presence of MPs in mussels from harvesting or production areas around the world. The mean values obtained in this study ranged from 0.05 to 4.6 MPs/g and from 1.5 to 7.6 MPs/individual, supporting the hypothesis that mussels are an important bioindicator of the presence of microplastics in the marine environment [1,20,21,24].

Recently, research focused on the potential risks associated with microplastic contamination of mussels and the potential implications for food safety and human health [7,25].

In our study, all samples tested were contaminated with MPs with average values of 1.59 ± 0.95 MPs/g and 6.51 ± 4.32 MPs/individual.

These results could be attributed to the areas of origin which could influence the level of contamination. According to the information on the labels, the samples tested came from Italy and Greece (FAO zone 37.2). It is known that the Mediterranean Sea is one of most affected by marine litter in the world due to its particular geography [26]. It’s estimated that more than 62 million items float on the surface of the Mediterranean Sea. The highest density of floating debris (>52 pieces/km^2^) was found in the Adriatic Sea and the Algerian Basin, while the lowest density (<6.3 pieces/km^2^) was observed in the central Tyrrhenian Sea and the Sicilian Sea [8]. In fact, the Mediterranean Sea is characterised by closed areas such as the Adriatic Sea, or semi-closed areas such as the Tyrrhenian Sea and the lower Mediterranean Sea. This feature could influence the different circulation of MPs and, consequently, their distribution in the different areas of the Mediterranean. Therefore, as a direct consequence, there would be a different concentration of MPs in bivalve molluscs, farmed or fished, marketed in the different Italian regions.

In our study, samples of mussels farmed in the lower Adriatic and Ionian Seas showed a lower incidence of MPs than the results of a study conducted in the upper Adriatic, where the number of MPs was higher (2 MPs/g and 12.4 MPs/individual) even compared to samples farmed and marketed in more open areas of the Mediterranean (Sardinia, La Spezia and Talamone) [17]. In a related study on *M. galloprovincialis*, *M. edulis*, *M. chilensis*, collected from fish markets and wholescalers in Sicily (Italy), MPs contamination was lower for the *M. galloprovincialis* from FAO area 37, but higher (0.39 ± 0.25 pieces/g wet weight) when compared to *M. edulis* and *M. chilensis* from other production areas (FAO area 27 and 87) [27].

Several other surveys confirm that the area of origin of mussels has an important influence on their contamination with MPs. Indeed, in samples of mussels (*M. galloprovincialis*), clams (*Ruditapes decussatus*) and oysters (*Crassostrea gigas*), farmed in the lagoon of Bizerte, Tunisia, (lower Mediterranean Sea), the average MPs value was 1.03 ± 3.5 MP/g (wet weight) for all bivalves considered [28]. Low MPs values were found in mussels farmed and marketed in Germany (0.36 ± 0.07 MP/g), in UK (0.9 MP/g) and South Korea (0.12 ± 0.11 MP/g) [1,16,29].

In addition, high MPs levels were recorded in China in a survey on nine samples of bivalve molluscs (from 4.3 to 57.2 MP/individual) and in Thailand on samples of green mussels (*Perna viridis*) (96 ± 19 MP/individual) [1,30].

MPs, which are ubiquitous in the marine environment, enter the food chain and affect commercially important fish species for human consumption. Several studies showed that microplastics in mussels reflect, for number, shape and size, the MPs in the waters surrounding the growth site of the molluscs.

### 4.2. MPs Shape, Colour and Size

Four types of MPs were identified in our study: fibres, fragments, plastic films and spherical granules. The first three types of MPs are of secondary origin and result from fragmentation or degradation of larger particles by mechanical forces and photochemical oxidation in the environment. The fourth type were spherical granules that can be classified as primary MPs and originate from ‘scrubbers’ used in the mechanical and cosmetic industries [3].

Fragments were the most prevalent type of MPs detected. This result is consistent with data of a survey performed on samples of mussels farmed in the northern Ionian Sea, and seawater samples from the same farming site [21]. Similarly, in a study on mussels sold in Korean markets, the number of MPs fragments was higher than that of other types (76%) [29].

The high incidence of fragments in mussels could be related to plastic sources and waste management strategies in the waters where they are fished and farmed. In the Mediterranean Sea, marine litter was detected on beaches, on the sea surface, in the water column and on the seabed [26]. They may originate from land-based anthropogenic activities (plastic bags, packaging materials or industrial waste) or from anthropogenic activities at sea (fishing gear and ship sewage).

Prolonged exposure to ultraviolet (UV) light and physical abrasion cause fragmentation into microplastics.

MPs enter the trophic chain of marine biota and humans because of their features, namely their wide colour range and small size. In addition, their similarity to prey increases the likelihood that they will be ingested.

Bivalves deserve a particular focus because their filter feeding mechanism let them filter high volume of seawater, thus accumulating large amount of MPs in the hepatopancreas and other tissues; therefore, they represent a route of exposure to MPs for humans, considering that they are consumed without gut removal and their consumption is high worldwide [1,29].

Mussel feeding strategies include capture, transport in the mucous layer and subsequent assimilation through the gills, mouth and digestive system. This mechanism is selective, in fact, mussels excrete larger MPs through faeces and retain smaller MPs (0–90 m) in tissues [31]. Due to their ability to accumulate MPs in the organism, mussels are considered an excellent bioindicator for monitoring and understanding the uptake, accumulation and toxicity of many anthropogenic pollutants, including microplastics [18,24].

### 4.3. Polymer Composition

The analysis of the MPs with FTIR-ATR suggested us that one of the main sources of microplastics analysed is the synthetic textile fibres used for the production of ropes, collectors, nets, hoses and buoys used in aquaculture facilities during the growth stages of mussels.

Therefore, one source of microplastics could be the tubular nets used for mussel farming, which are not replaced during their growth and remain attached to the mussels.

PP was one of the polymers identified by FTIR-ATR, but to a lesser extent than the other polymers (Table 2), despite being one of the most widely used plastics in the world for food packaging, candy and snack wrappers, microwave containers, tubes and more [32]. In fact, in other studies PP, PE, PET (including polyester) and PA were the polymers most identified in bivalve molluscs [1,18,28,29,32].

Furthermore, farming methods (aquaculture equipment and water depth) and physical and chemical properties of microplastics (specific density and shape) influence the type of microplastics found in fish products [29,33]. For example, polymers with low specific density such as PE, PP (in a range of 0.85 to 0.98 g/cm^3^) and EPS (0.015 g/cm^3^), are concentrated in surface waters and consequently on beaches, while high specific density polymers such as polyester (1.3 g/cm^3^) are concentrated in the lower water column and sediments [5].

Therefore, polyamide, due to its density (1.12 g/cm^3^), was the most isolated type of polymers in our study, as they were found to concentrate in the middle layer of the water column where the mussels were reared. In aquaculture facilities, oysters and mussels are cultured at a depth between 3 and 8 m in the water column, while scallops are cultured in the middle or lower layer below 15 m and demersal molluscs such as clams are cultured in the intertidal zone [29].

However, chemical identification of polymers is not always possible. One reason is the degradation of microplastics due to the use of overly aggressive solutions to isolate them, the second reason is the presence of non-synthetic fibres that cannot be chemically identified but are nevertheless included in the enumeration. The immediate consequence is an over- or underestimation of the concentration of MPs.

### 4.4. MPs Detection, Isolation and Identification Protocols

To date, there is no standardised protocol for monitoring and assessing of MPs in food, or a guideline for sampling, detection and identification of microplastics that allows reproducible and repeatable results. Several protocols are described for detection, isolation and identification of microplastics in food and environment, and for this reason, results are not consistent.

For example, the use of KOH or HNO_3_ for soft tissue digestion gave good results, and eased the isolation of plastic particles. However, these solutions can lead to the degradation of polymers such as polyamide, polystyrene and polyethylene, as they are too basic or too acidic. Therefore, the use of overly aggressive reagents could affect surface, colour and size of some types of MPs, reducing extraction efficiency and leading to underestimation of results [34].

The protocol used in our study gave results consistent with those reported in the literature. In detail, the use of H_2_O_2_ completely degraded the organic matter and did not produce foam during digestion, which would have affected the detection of MPs microscopically. In addition, the use of hypersaline NaCl solution (1.2 g/cm^3^) for the separation of polymers based on the density gradient, allowed a better recovery of MPs from organic matter. Finally, the microscopic observation of the samples and the subsequent chemical identification of the polymers by FTIR-ATR allowed the presence of MPs not to be overestimated.

### 4.5. MPs and Risks Associated with Commercialisation of Mussels

In the European Union (EU), bivalve molluscs must comply with the microbiological criteria of Reg. EC 2073/2005 and, if necessary, undergo a depuration period before being marketed.

A recent study showed that environmental factors, such as the presence of shellfish feed in the marine environment, can influence the fate of MPs. The authors claimed that mussels excrete MPs more slowly when ingested at the same time as beneficial, nutritious algal cells. In contrast, MPs are excreted rapidly when nutrients are not present [35]. This might suggest that a nutrient-deprived environmental system, as occurs during the depuration process, would ease the elimination of MPs. Indeed, prolonged depuration may lead to the expulsion of mainly larger particles, while smaller particles may be transferred to tissues through the gastrointestinal tract and circulatory system [16].

According to Sobhani et al. [35], fresh and processed bivalve molluscs are subject to handling at all stages of the food chain (from production to home consumption), thus exposing them to contamination with MPs. Indeed, it was shown that the opening of packaging or the handling of plastic films can generate microplastics and their quantity is influenced by the force applied in the opening of packaging and by the type of plastic. Consequently, it can be assumed that contamination of bivalve molluscs can occur during all stages of processing, including during the sale.

During the commercialization, MPs contamination of bivalve molluscs may increase due to some common habits such as dipping them in water or, at the consumer’s request, shucking and packing them in plastic packages or bags [36].

### 4.6. MPs and Risks to Consumers Health

Shellfish consumption is an important route of exposure to MPs for consumers. The estimated daily intake of MPs through shellfish consumption was calculated based on the amount of product and MPs concentrations found in different species. The potential risks to human health from MPs ingestion are still unclear; however, the role of MPs as a carrier of drugs, chemical contaminants and microbial communities that may find an ideal habitat on the surface, should not be ignored [37].

The biofilms produced by bacteria colonising the surface of plastics could become a reservoir for pathogenic bacteria, faecal indicator organisms and algal bloom species, creating a microcosm in which antibiotic resistance can be transferred to other bacteria that lack it [38,39]. Therefore, human ingestion of MPs through consumption of contaminated bivalve molluscs can pose a serious risk to human health. In order to understand when exposure to MPs and contamination of bivalve molluscs is greatest, further investigations are needed across all stages of the food chain, from primary production to commercialization.

Furthermore, to assess the potential impact of MPs on the quality and safety of mussels and other bivalve molluscs, analytical methods for the detection and quantification of MPs must be standardised to ensure a proper interpretation of the results. In order to perform a proper risk analysis, the whole bivalve molluscs chain must be investigated, starting from the monitoring of production and harvesting sites, to the processing, packaging and marketing operations. There is growing evidence that the global pollution of the marine environment by plastic waste has strong implications for food safety and human health; the presence of microplastics in commercially harvested mussels poses a potential risk to consumers, but there is hope for global regulatory solutions that will lead to better management of plastic waste.

## 5. Conclusions

This market-based study provided evidence of the occurrence of microplastics in fresh and live mussels (*M. galloprovincialis*) sold in local fishing stores in Bari and its neighbouring cities in the Apulia region (Italy).

Our results showed a contamination of mussels that reflected the pollution level of the waters where they were harvested or farmed. Nevertheless, contamination may occur during commercial distribution, due to plastic packaging or inappropriate storage conditions, thus affecting the abundance of plastic micro-particles in bivalves. In any case, it must be considered that the lack of standardized protocols and the use of overly aggressive solutions, such as KOH, may also lead to an over- or underestimation of contamination.

Entering the food chain through diet, mussels represent a source of microplastics for consumers and a direct risk to their health, as they are carriers of many chemical compounds and microorganisms that may or not be pathogenic to human health.

FTIR-ATR allowed to detect a reliable chemical identity of the polymers of our plastic findings, such as Polyamide, suggesting that the main source of contamination could be related to the ropes and gears used in aquaculture, but contamination at the marketing stage cannot be excluded. Further studies are needed to define analytical standardized protocols for the isolation of microplastics, and to assess the impact of commercialization on the occurrence of MPs in bivalve mollusc, in order to protect consumer health.

## Figures and Tables

**Figure 1 foods-12-01495-f001:**
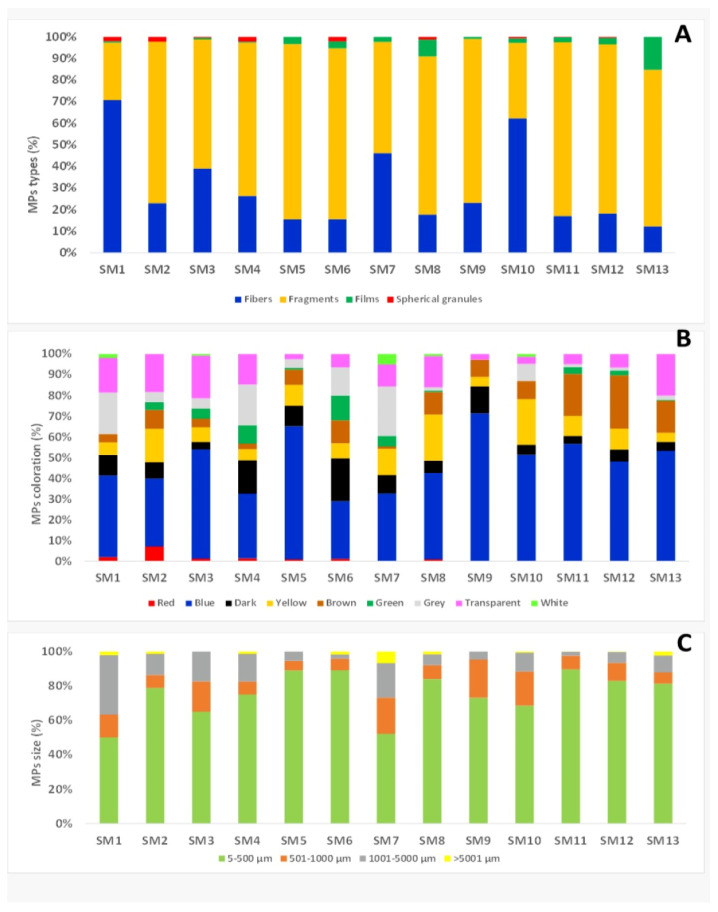
Distribution by type (**A**), colour (**B**) and size categories (**C**) of MPs in mussel samples.

**Figure 2 foods-12-01495-f002:**
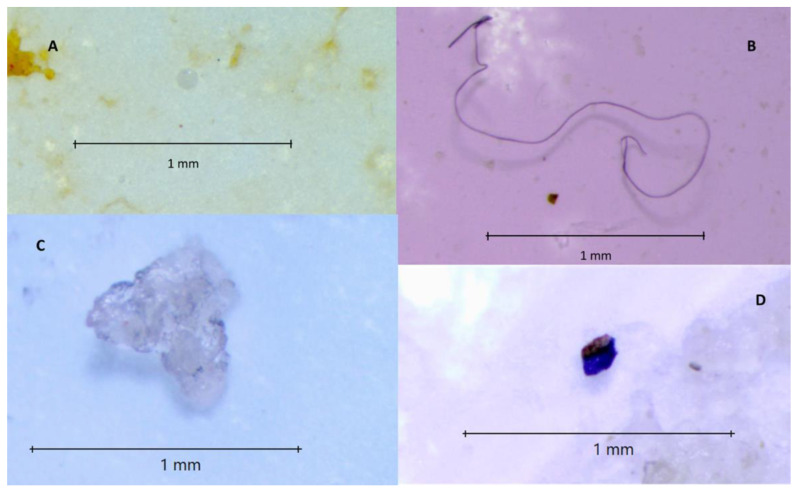
MPs isolated from mussel samples. (**A**) spherical granule; (**B**) fibre; (**C**) film; (**D**) fragment.

**Figure 3 foods-12-01495-f003:**
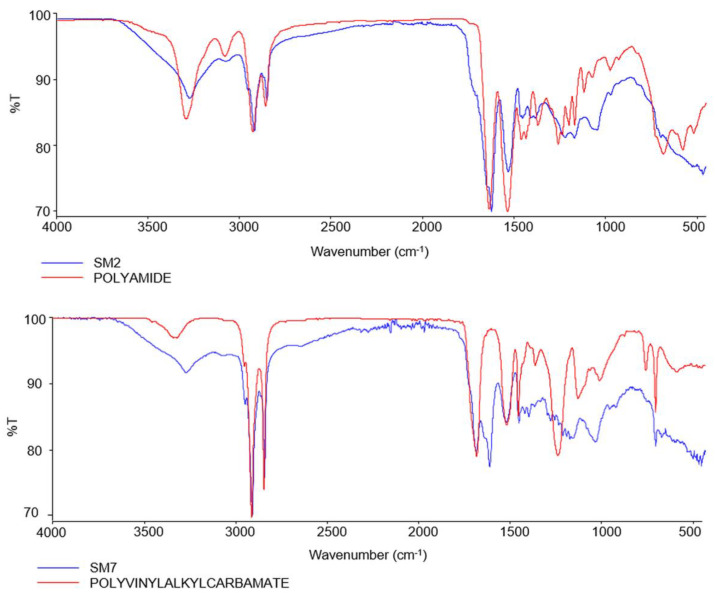
Spectra resulting from FTIR-ATR analysis of samples SM2 and SM7.

**Table 1 foods-12-01495-t001:** Average values of MPs in the three repetitions for each mussel sample.

Sample	Shell Length (cm)	Soft Tissue Weight (g/Total)	n. MPs	n. MP/g	n. MP/Individual
SM1	7.2 ± 0.7	255	150	0.6 ± 0.07	2.5 ± 0.3
SM2	6.3 ± 0.3	204	396	1.9 ± 0.18	6.6 ± 0.62
SM3	5.5 ±1.4	195	243	1.25 ± 0.09	4.1 ± 0.28
SM4	6.4 ± 1.0	240	224	0.93 ± 0.06	3.7 ± 0.24
SM5	6.7 ± 1.2	250	356	1.43 ± 0.12	5.9 ± 0.51
SM6	7.2 ± 0.4	265	431	1.66 ± 0.15	7.3 ± 0.8
SM7	6.2 ± 1.9	210	260	1.24 ± 0.08	4.3 ± 0.29
SM8	7.5 ± 0.6	270	876	3.24 ± 0.07	14,6 ± 0.64
SM9	6.9 ± 0.8	205	108	0.53 ± 0.21	1.8 ± 0.74
SM10	7.1 ± 0.4	263	146	0.55 ± 0.24	2.43 ± 1.12
SM11	5.7 ± 1.8	188	924	3.62 ± 0.88	15.4 ± 3.75
SM12	7.2 ± 0.5	248	516	2.02 ± 0.10	8.6 ± 0.44
SM13	6.9 ± 1.1	215	447	1.75 ± 0.16	7.45 ± 0.68
Average values	6.68 ± 0.61	231 ± 29.23	390.54 ± 259.18	1.59 ± 0.95	6.51 ± 4.32
Total			5077		

**Table 2 foods-12-01495-t002:** Polymers identified by FTIR-ATR in mussel samples.

Sample	Polymers	%
SM1	PA	69.23
SM2		
SM3		
SM4		
SM5		
SM6		
SM7		
SM8		
SM9		
SM10		
SM11		
SM12		
SM13		
SM7	PVAC	11.53
SM5		
SM6	PP	7.69
SM8		
SM3	PUR	11.53
SM11		
SM12		

Legend: PA: Polyamide; PVAC: Polyvinylalkylcarbamate; PP: Polypropylene; PUR: Polyurethane.

## Data Availability

The data are available from the corresponding author.

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
