# Peer review of "Occurrence and Characterization of Microplastics in Commercial Mussels (Mytilus galloprovincialis) from Apulia Region (Italy)"

_foods, 2023, doi:10.3390/foods12071495_

Round 1
Reviewer 1 Report
1. Lines 132,141, 295, 296, 395, 402 the numbers in formula should be in superscript or subscript.
2. The authors used 1.2 g/ml NaOH solution to separate MPs via floating. However, the density of some plastics materials is higher than this value (e.g. PET, PVC, PTFE) and thus this MPs will not float but sediment. Did you start with assumption that PET, PVC.. particles will fall to the sea bottom or deeper in the sea than the depth where mussels are located?
3.lines 185-186: the airborne MPs is 10% of the maximum value obtained for sample SM11. Isn’t that high?
4. the text in Figure 1 should be larger since it is very tiny and hard to read.
5. Nylon 6,6 is polyamide! Polyamide is a group of polymer materials, therefore you cannot talk about nylon 6,6 and polyamide as different categories of polymers, you can only identify nylon 6,6 as a subcategory of polyamides. Correct Figure 2 and Table 2 accordingly. As seen in Figure 2, red curves in both spectra are identical since Nylon 6,6 is polyamide. Also, add numbers to the Y-axe.
6. The reviewer suggests to analyze the material of found spherical granules, fibers, films and fragments to determine if they are polymers and if they are, to identify them.
7. lines 324-326: revise the sentence: “Several studies…their growth.”
8. Lines 371-372: revise the sentence” In fact…PA.”
9. Line 376:PE and PP have specific densities in a range of 0.85 to 0.98 g/cm3, whereas the density of EPS is 0.015 g/cm3
10. The reviewer suggests to take into account salinity of the sea, as well as the temperature and depth where the mussels where taken from, since all this factor affect the sea density.
Author Response
"Please see the attachment.

Reviewer 2 Report
This study investigated MPs contamination in edible mussels. The topic is interesting for the reader of this journal. The results were significant. But the presentation is of low quality. The description of the analysis is insufficient.
In my experience, FTIR-ATR requires sample size larger than 5 mm in diameter. Did you use some special equipments to analyze the MPs (< 1mm, Fig. 3)?
The structure of the sections and paragraphs was strange. Research papers should be well organized structured and separated by sections. The section should be separated by paragraphs. Each first sentence of the paragraphs should represent the topic of each paragraph. Well organized section has two rules. First, one paragraph can describe only one topic. Second, the first sentences should be an abstract of the section.
For example,
Section A
First paragraph: First sentence (1). Following sentences.
Second paragraph: First sentence (2). Following sentences.
Third paragraph: First sentence (3). Following sentences.
Fourth paragraph: First sentence (4). Following sentences.
Abstract of Section A: First sentence (1). First sentence (2). First sentence (3). First sentence (4).
Please re-organize the manuscript based on the rules.
Author Response
"Please see the attachment.

Round 2
Reviewer 2 Report
You can obtain books elsewhere to learn how to structure paragraphs, as you can search 'paragraph writing' by Google.
